# Characterizing Methicillin-Resistant *Staphylococcus* spp. and Extended-Spectrum Cephalosporin-Resistant *Escherichia coli* in Cattle

**DOI:** 10.3390/ani14233383

**Published:** 2024-11-25

**Authors:** Lisa Abdank, Igor Loncaric, Sascha D. Braun, Elke Müller, Stefan Monecke, Ralf Ehricht, Reinhild Krametter-Frötscher

**Affiliations:** 1Clinical Centre for Ruminant and Camelid Medicine, University of Veterinary Medicine, 1210 Vienna, Austria; reinhild.krametter@vetmeduni.ac.at; 2Institute of Microbiology, University of Veterinary Medicine, 1210 Vienna, Austria; igor.loncaric@vetmeduni.ac.at; 3Leibniz Institute of Photonic Technology (IPHT), Leibniz Center for Photonics in Infection Research (LPI), 07745 Jena, Germany; sascha.braun@leibniz-ipht.de (S.D.B.); elke.mueller@leibniz-ipht.de (E.M.); stefan.monecke@leibniz-ipht.de (S.M.); ralf.ehricht@leibniz-ipht.de (R.E.); 4InfectoGnostics Research Campus, 07743 Jena, Germany; 5Institute of Physical Chemistry, Friedrich-Schiller University, 07743 Jena, Germany

**Keywords:** MRSA, ESBL, AmpC, cattle, bovine, *E. coli*, AMR, mastitis, pneumonia, diarrhea, virulence genes

## Abstract

The role of methicillin-resistant *Staphylococcus* (*S*.) *aureus* (MRSA) and broad-spectrum cephalosporin-resistant *Escherichia* (*E*.) *coli* in cattle has not yet been widely investigated in Austria. This study aimed to understand the presence of these bacteria in certain regions of Lower Austria. A total of 190 milk samples from cows and 123 nasal swabs from cattle were examined for the presence of MRSA, as well as 99 bovine fecal swabs for *E. coli*. The samples were taken from 66 participating farms between May 2021 and September 2022 as part of the veterinary herd monitoring program of a veterinary practice in Lower Austria. MRSA was detected in a single nasal swab, with no MRSA found in the milk samples. A total of 22 *E. coli* isolates (22.2%) were detected and displayed an extended-spectrum β-lactamase (ESBL) phenotype. One *E. coli* isolate also harbored the AmpC gene. Finally, the isolates were analyzed for the following microbiological tests: DNA microarray, PCRs, and *spa* typing. The results conclusively showed that antibiotic resistance does play a role in cattle in (Lower) Austria.

## 1. Introduction

In veterinary medicine, methicillin-resistant *Staphylococcus aureus* (MRSA) and extended-spectrum β-lactamase (ESBL)-producing *Escherichia coli* are considered challenging pathogens to treat due to increasing antimicrobial resistance (AMR). Both bacteria are significant pathogens responsible for various infectious diseases in humans and animals [1,2,3].

*E. coli* is a multifaceted bacterium and part of the normal microbiota but can also play an important role as a pathogen. Some *E. coli* strains harbor virulence-associated genes (VAGs), which can be responsible for different diseases in humans and animals [4]. In calves, *E. coli* causes acute diarrhea that can lead to severe clinical symptoms and death without the appropriate treatment [3,5,6]. Depending on VAGs, pathomechanisms, and clinical symptoms, *E. coli* strains are categorized into several pathotypes. Diarrhea-associated strains involve enteropathogenic *E. coli* (EPEC), enteroaggregative *E. coli* (EAEC), enterotoxigenic *E. coli* (ETEC), enteroinvasive *E. coli* (EIEC), and enterohemorrhagic *E. coli* (EHEC). Extraintestinal infections are caused by extraintestinal pathogenic *E. coli* strains (ExPEC). ExPEC are mostly harmless intestinal commensals and are only harmful if they reach, or are displaced to, other parts of the body [7,8,9]. *E. coli* has acquired resistance mechanisms and a genetic adaptation to exposure to antibiotics. Due to this genetic adaptation, antimicrobials may be less effective against *E. coli*, which can result in reduced susceptibility to antimicrobials [10,11]. *E. coli* can highly accumulate resistance genes, mostly through horizontal gene transfer, which means that this bacterial species is no longer susceptible to antimicrobial agents. The most problematic mechanisms in *E. coli* correspond to the acquisition of genes encoding carbapenemases, extended-spectrum β-lactamases, plasmid-mediated quinolone resistance genes, 16S rRNA methylases, and mcr genes [12]. ESBL genes among *E. coli* from animals are associated with several insertion sequences (ISs; ISEcp1, ISCR1, IS26, IS10) transposons (Tn2), and integrons [12,13]. Most ESBL genes are located in plasmids, and the most prevalent replicon types identified among ESBL-carrying plasmids from *E. coli* are IncF, IncI1, IncN, IncHI1, and IncHI2 [12]. Some plasmids harbor other resistance genes apart from the ESBL gene, which may facilitate the coselection and persistence of ESBL gene-carrying plasmids, even without the selection pressure of β-lactams, when the appropriate antimicrobial agents are used [12,14].

AmpC-β-lactamases are clinically important cephalosporinases and are encoded chromosomes of many Enterobacterales, where they mediate the resistance to cefoxitin, cefazolin, cephalothin, most penicillins, and β-lactam combinations [15]. AmpC-β-lactamase from *E. coli* was the first bacterial enzyme to destroy penicillin. Mutations with progressively increased penicillin resistance were termed *ampA* and *ampB*. A mutation in an *ampA* strain that resulted in decreased resistance was finally termed *ampC*, making little, if any, β-lactamase. Transmissible plasmids have acquired genes for AmpC enzymes, which can appear in bacteria poorly expressing the chromosomal *bla*_AmpC_ gene (such as *E. coli*). Such resistances are less common than ESBL production but can be more difficult to detect and have a broader spectrum [15].

Staphylococci are part of the physiological microbiota of the skin and mucous membranes of humans and different animals, and they are frequently associated with opportunistic infections [16,17]. In cows, *S. aureus* is a very relevant pathogen in udder infections, but it also can be found on the skin and mucous membranes, such as the nasopharynx [18,19,20,21]. Their effects are often exacerbated by the extended AMR of affected isolates. Methicillin-resistant *Staphylococci*, particularly *S. aureus*, are a major cause of nosocomial infections and life-threatening syndromes worldwide. *S. aureus* can exhibit resistance to various antibiotics, especially β-lactam antibiotics. Resistance to β-lactam antibiotics is caused by modified penicillin-binding proteins, PBP2a or PBP2c-, which have only a low affinity for β-lactams and involve, respectively, the *mecA* and *mecC* genes. Until now, the following different *mec* genes have been known to occur in *S. aureus*: *mecA*, *mecB*, and *mecC*. The genes *mecA* and *mecC* are located on chromosomal but potentially mobile genetic elements named the Staphylococcal Cassette Chromosome *mec* (SCC*mec*) and encode PBP2a [22,23]. It is assumed that *mecA* has long been present in Staphylococci other than *S. aureus*. It can therefore be assumed that different alleles of this gene are detectable and that some of them could play a role other than the transmission of antibiotic resistance. Consequently, this possible diversity of *mecA* alleles could be of great practical importance for the development of tests to confirm or detect *mecA* as a marker for methicillin/lactam resistance in routine clinical diagnostics [24]. The *mecB* gene (originally designated as *mecA_m_*) is part of a methicillin resistance gene complex [25]. It has been detected on a multiresistance plasmid in a human cefoxitin-resistant *S. aureus* isolate [26]. Shore et al. [27] and García-Álvarez et al. [28] were the first to identify the *mecC* gene (a strongly deviating *mecA* gene) and described the gene encoding *mecC* in MRSA as potentially zoonotically transmissible. Surveys of hedgehogs from Denmark and Sweden revealed a high prevalence of MRSA-carrying *mecC* (*mecC*-MRSA) in these animals [29,30].

The general aim of this study was to investigate the presence of ESBL-producing *E. coli* and MRSA in cattle in Austrian stables and to what extent AMR plays a role in the presence of these bacteria.

## 2. Materials and Methods

The samples collected were taken from the districts of Wiener Neustadt/Land, Neunkirchen, and the Wechsel Region. A total of 412 animals were sampled. Of these, 190 samples were taken from dairy cows with mastitis and 222 samples from calves, with 123 calves suffering from pneumonia and 99 from diarrhea. Overall, the samples originated from 66 different cattle herds. Sterile swabs (Transwab^®^ M40 Compliant, MWE, London, UK) were used for fecal samples and nasal swabs. Sterile centrifuge tubes were used for sterile milk samples (Covetrus AT GmbH, Brunn am Gebirge, Lower Austria, Austria).

### 2.1. Sample Collection

For the cultivation of fecal samples, they were first incubated overnight at 37 °C in buffered peptone water (Merck, Rahway, NJ, USA) with cefotaxime (1 mg/L) and subsequently cultured overnight at 37 °C on MacConkey agar (Oxoid; Basingstoke, UK) supplemented with cefotaxime (1 mg/L) (MacCTX). After the incubation on MacCTX, colonies which presented a specific *E. coli* colony morphotype were subcultured on the same medium and then cryo-conserved. Selection of the colony was based on form (e.g., circular, irregular), elevation (e.g., flat, convex), margin (e.g., entire, undulate), and mucoid vs. non-mucoid. Matrix-assisted laser desorption ionization time-of-flight mass spectrometry (MALDI-TOF MS) (Bruker Daltonik; Bremen, Germany) was used to identify the isolates to the species level. Only isolates that were confirmed as *E. coli* were selected for further characterization. Antimicrobial susceptibility testing was performed by agar disk diffusion according to the CLSI standards [31]. *E. coli* ATCC^®^ 25922 served as the quality control. Disks containing the following antimicrobial agents were used: cefotaxime (30 µg); ceftazidime (30 µg); cefoxitin (30 µg); meropenem (10 µg); gentamicin (10 µg); tobramycin (10 µg); amikacin (30 µg); ciprofloxacin (5 µg); trimethoprim–sulfamethoxazole (1.25/23.75 µg); tetracycline (30 µg); chloramphenicol (30 µg); fosfomycin (200 µg); and nitrofurantoin (300 µg) (Becton Dickinson; Heidelberg, Germany). Resistance and virulence genes were analyzed by INTER-ARRAY Genotyping Kit CarbaResist (INTER-ARRAY by fzmb GmbH; Bad Langensalza, Germany) [32] as well as by PCR (i.e.; *catA*; *cmlA*; *floR*; *tet*(A), *tet*(B)) as described elsewhere [33]. Detection and analysis of virulence-associated genes were performed using custom-made microarrays from INTER-ARRAY (INTER-ARRAY by fzmb GmbH, Bad Langensalza, Germany) according to the manufacturer’s instructions [34]. The phylogroup of the *E. coli* isolates was determined by the revisited Clermont method [35]. DNA microarray results were visualized as previously described [33], and the program SplitsTree4 (SplitsTree_windows-x64_6_3_33.exe) on default settings [36] was used.

### 2.2. Isolation of MRSA, Antimicrobial Susceptibility Testing

The nasal swabs were incubated overnight in tryptic soy broth (Beckton Dickinson (BD); Heidelberg, Germany) with 6.5% (*w*/*v*) NaCl and then incubated on BBL™ CHROMagar™ MRSA II (BD). The *S. aureus* colonies that showed the typical colony pattern of MRSA after incubation on BBL™ CHROMagar™ MRSA II were selected. Cefoxitin resistance was confirmed by agar disk diffusion [31,37]. Agar disk diffusion was performed according to CLSI document M100 (28th ed.) [33]. The following antimicrobial agents have been tested: gentamicin (GEN, 10 μg), erythromycin (ERY, 15 μg), penicillin (PEN, 10 IU), ciprofloxacin (CIP, 5 μg), clindamycin (CLI, 2 μg), tetracycline (TET, 30 μg), trimethoprim–sulfamethoxazole (SXT, 1.25/23.75 μg), chloramphenicol (CHL, 30 μg), and linezolid (LZD, 30 μg). The reference strain *S. aureus* ATCC^®^ 29523 served as a quality control strain. 

The milk samples were cultivated and identified as previously described by Keinprecht et al. [38]. 

### 2.3. Detection of Antimicrobial Resistance Genes and Molecular Characterization of MRSA

Microarray-based detection of the virulence-associated genes was performed as described by Bernreiter-Hofer et al. [34]. Genotyping by Inter-Array Kit CarbaResist and the Microarray Hybridization Kit were performed as described by Bedenic et al. [39], Braun et al. [32,40,41], and Monecke et al. [42]. Detailed information about all target genes from the Inter-Array is available in the Appendix A. DNA extraction was carried out as previously described by Loncaric et al. [43]. *Spa* typing was performed as previously described by Loncaric et al. [43]. 

Resistance genes were analyzed by PCR (i.e.; *catA*; *cmlA*; *floR*; *tet*(*A*), *tet*(*B*)) as described elsewhere [33].

## 3. Results

### 3.1. Antimicrobial Susceptibility Testing and Characterization of Genotypic Antibiotic Resistance of E. coli

A total of 22 *E. coli* isolates were grown on MacConkey agar with cefotaxime (MacCTX, Rapid Labs, Colchester, UK) and identified as *E. coli* by MALDI-TOF MS. Out of the 412 samples, 99 (24.0%) came from calves with diarrhea. Of these, 21 (21.2%) positive samples and 22 (22.2%) isolates were detected.

All isolates were susceptible to carbapenems, amikacin, tobramycin, nitrofurantoin, and colistin, and displayed an ESBL phenotype including resistance to ceftazidime. Out of the 22 *E. coli* isolates, all the isolates were resistant to at least one non-β-lactam antibiotic tested. In total, 63.6% (n = 14) of the isolates exhibited a multidrug-resistant (MDR) phenotype [44]. Resistance to β-lactams, trimethoprim–sulfamethoxazole (SXT), and tetracyclines were most common among the multidrug-resistant phenotypes (n = 13, 59.1%). Further, the most frequently observed resistance property was combined resistance to β-lactam antibiotics and trimethoprim-sulfamethoxazole (n = 21, 95.5%). Concerning antibiotic resistance to non-β-lactam antibiotics, the following results were obtained: antibiotic resistance to trimethoprim–sulfamethoxazole (n = 21, 95.5%), tetracyclines (n = 13, 59.1%), ciprofloxacin (n = 7, 31.8%), chloramphenicol (n = 7, 31.8%), and gentamicin (n = 3, 13.6%) was determined. 

In 100% of the isolates, genes from the *bla*_CTX_ family were detected with other *bla* genes or alone. The most frequently observed β-lactamase genes were *bla*_CTX_ genes, *bla*_CTX-M-1/15_ (n = 20, 90.9%), and *bla*_CTX-M9_ (n = 2, 9.1%), followed by *bla*_TEM_ (n = 18, 81.8%). Concerning the resistance to tetracyclines, it was mediated by the *tet*(A) (n = 13, 59.1%) and the *tet*(B) (n = 2, 9.1%) genes. Regarding trimethoprim–sulfamethoxazole resistance, 17 isolates (77.3%) carried *sul2* genes and 10 isolates (45.5%) carried *dfrA14* genes. Further, the following genes were detected by decreasing order: *dfrA5* (n = 8, 36.4%), *dfrA1* (n = 6, 27.3%), *dfrA17* (n = 3, 13.6%), *sul3* (n = 2, 9.1%), and *sul1* (n = 2, 9.1%) (Table 1 and Appendix A). Additionally, one *E. coli* isolate (4.5%) harbored the AmpC gene (*bla*_ACT_) and showed a reduction in the inhibition zone of cefoxitin.

### 3.2. Virulence Associated Genes and Phylotype of E. coli

The type I fimbrial protein *fimH1* was most frequently detected; all the 22 isolates were tested positive for this VAG, followed closely by *fimH2*, which occurred 19 times (86.4%). Concerning the extraintestinal VAGs, such as *papC* and *iucD*, the following results were obtained: the outer membrane usher P fimbriae *papC*, as well as the aerobactin biosynthesis proteins *iucD*, occurred six times (27.7%) each. The gene encoding virulence factor for Hemolysin *hlyA* was detected in two isolates (9.1%). 

Among all *E. coli* isolates, the predominant phylogenetic group was A (50%), followed by B1 (45.5%). The one remaining isolate belonged to the E clades (4.5%).

A SplitsTree analysis of the microarray data revealed clonal clustering into 14 groups based on their similarities in virulence and antimicrobial resistance profiles (Figure 1).

### 3.3. MRSA Characterization

Out of the 412 samples, 190 (46.1%) were from cows suffering from mastitis, and 123 (29.9%) were from pneumonia calves. No positive sample was detected in the 190 milk samples. Of the 123 samples from calves suffering from pneumonia, 1 (0.8%) MRSA isolate was found. 

The MRSA isolate was *mecA* positive and belonged to *spa* type t011 and clonal complex (CC) 398. The *mecA* gene was carried on SCC*mec* type IV. Aside from β-lactam resistance, this isolate was resistant to gentamicin and tetracycline, which was in accordance with the observation that this isolate carried resistance genes *aacA-aphD* and *tet*(M). The gamma-hemolysin locus genes *lukF-hlg*, *lukS-hlg*, and *hlgA* were detected; the bovine-associated leukocidin genes *lukM/lukF-P83* were absent. The MRSA isolate harbored the hemolysin *hla* gene.

## 4. Discussion

This study aimed to determine the presence of extended-spectrum cephalosporin-resistant *E. coli* in fecal samples from calves as well as MRSA in nasal swabs from calves and milk samples from cows in certain regions in Lower Austria. Additionally, the study included a characterization of the *E. coli* isolates and the MRSA isolate. 

As previously described, all the 22 *E. coli* isolates displayed an ESBL phenotype. Comparable Austrian studies for cattle regarding *E. coli* isolates carrying an ESBL phenotype and genotype are scarce. In a similar, comparable study from Austria, 138 fecal samples were taken from 50 dairy farms to test for ESBL-producing *E. coli* from cow and calf stables as well as from youngstock housing areas [45]. The results showed that a total of 13 (26%) of the 50 participating farms were positive for the presence of ESBL-producing *E. coli*. More detailed information on the phenotype and genotype was not listed in the survey [45]. The difference between this study and the study from Lower Austria is that the results regarding ESBL-producing *E. coli* from Lower Austria only came from calves. At the same time, the other Austrian study also took fecal samples from cows and young cattle. A comparable investigation from neighboring Germany examined extended-spectrum β-lactamase/plasmid-mediated AmpC β-lactamase-producing *E. coli* isolates from livestock farms and found that 68% of the isolates from cattle carried *bla*_CTX-M_ genes [46]. A similar result is reflected in the current study, where *bla*_CTX-M_ genes (*bla*_CTX-M-1/15_, n = 20, 90.9%) are also detected most frequently. The present study shows consistent results with that of the German study, with phylogenetic group A being the most frequently detected of the *E. coli* isolates, at 50% and 55%, respectively [46]. The following, comparable studies from France and Switzerland also list related results, where they detected *bla*_CTX-M_ genes and the phylogenetic group A most frequently in fecal samples from cattle. The French study analyzed 204 ESBL-producing *E. coli* isolates from diarrheic cattle, with the results that ESBL genes belonged mostly to the *bla*_CTX-M-1_ (65.7%) and *bla*_CTX-M-9_ (27.0%) groups, the dominant phylogenetic group was phylogroup A (55.4%), and phylogenetic group B1 had a lower ratio of 15.6% [47]. In the Swiss study, a total of 196 fecal samples were taken from calves, and 18 ESBL-producing *E. coli* were isolated. Of these, eight isolates carried the *bla*_CTX-M-15_ gene, four isolates harbored *bla*_CTX-M-1_, four isolates harbored *bla*_CTX-M-3_, and two *E. coli* isolates contained *bla*_CTX-M-14_ [48]. The phylogenetic classification showed that 13 *E. coli* isolates were assigned to phylogenetic group A, 4 to phylogenetic group B1, and 1 strain to group C [48]. The two studies [33,38,47,48] differ in the following respect: In the French study, as in the one from Lower Austria, fecal samples were taken from calves with diarrhea, while in Switzerland, healthy calves were sampled. Concerning animal groups and the phylotype, a study from the USA examined *E. coli* isolates from humans and various animals about phylogenetic groups, genotypic clusters, and virulence gene profiles [49]. In ruminants, namely cows, goats, and sheep, a majority of the *E. coli* isolates belonged to phylogenetic group B1 [49]. In an older study from the USA, which also investigated the occurrence of phylogenetic groups in different animal species, it was reported that group B1 was dominant in cows as well [50]. The limiting factor for comparing this study with the others is that triplex PCR was used, and therefore not all the phylogenetic groups were included. These results are also reflected in a study from 2010, which detected 241 *E. coli* strains isolated from the feces of different animals and humans, that the phylogenetic group B1 was more prevalent in cow, goat, and sheep samples [51]. Another study from Poland examined 300 *E*. *coli* isolates from herbivorous, carnivorous, and omnivorous mammals from a zoo, which were characterized for their phylogenetic origin, intestinal virulence gene prevalence, and genomic diversity [52]. The phylogenetic structure of the *E. coli* from the herbivores (aurochs, buffalo, eland, waterbuck, yak) showed group B1 with a prevailing representation. In omnivores (dingo, raccoon) and carnivores (lion, lynx, wildcat), group A showed a higher representation in comparison to the herbivores [52]. These studies [49,50,51,52] show that phylogenetic group B1 is the dominant group in ruminants and not phylogenetic group A, as in the study from Lower Austria and the other studies previously mentioned [46,47,48]. This demonstrates that it is probably impossible to generalize the predominant phylogenetic group in cattle or ruminants worldwide, but rather only a possible tendency.

Regarding the AmpC-genes in this study, one *E. coli* isolate harbored an AmpC-gene. A comparable investigation from neighboring Germany, which examined extended-spectrum β-lactamase/plasmid-mediated AmpC β-lactamase-producing *E. coli* isolates from livestock farms showed similar results [46]. Plasmid-mediated AmpC (pAmpC), was not detected in dairy cattle in the German study and less than 5% of the samples from beef cattle (n = 82) tested positive for pAmpC. Another comparable study—also from Germany—determined the prevalence of ESBL/AmpC-producing *E. coli* from 120 fecal samples in dairy farms from cattle of all ages [53]. The DNA sequence analysis revealed that all isolates carried AmpC1 and of the 20 samples obtained from calves, 100% displayed phenotypic cefotaxime resistance [53]. Even though only one positive *E. coli* isolate carried an AmpC gene in the study from Lower Austria, numerous studies on ESBL/pAmpC β-lactamase-producing *E. coli* in livestock animals have been published in recent years [15,53], but comparative data for Austria are scarce.

Concerning the *E. coli* isolates from the present study, all isolates exhibited resistance against penicillins and cephalosporins tested, 21 (95.5%) showed antimicrobial resistance to trimethoprim–sulfamethoxazole. Another Austrian study from 2020 and 2022 investigated the prevalence of ESBL-producing *E. coli* on dairy farms of fecal samples from calves and cows [54]. In total, 14 (n = 198, 7.1%) ESBL-producing *E. coli* were detected in 2020 and 41 (n = 190, 21.6%) in 2022. All ESBL-producing *E. coli* isolates were susceptible to polypeptides, carbapenems, and tigecycline [54], compared to the isolates from the present study in lower Austria, which were also all susceptible to carbapenems and colistin. The difference to the investigation from Lower Austria is that not only calves but also cows were sampled [54]. A German study was designed to assess the prevalence of ESBL-producing *E. coli* in cattle with the result that out of 598 samples, 196 (32.8%) contained ESBL-producing *E. coli* [55]. In comparison with the present study, it was shown that out of 99 fecal samples, 21 (21.2%) contained ESBL-producing *E. coli*. All isolates from Germany showed susceptibility to carbapenems (except for one ertapenem-resistant isolate). This German study has also shown that ESBL-producing *E. coli* from cattle farms were significantly less resistant to aztreonam, gentamicin, tobramycin, ciprofloxacin, and trimethoprim–sulfamethoxazole than isolates from mixed farms. Concerning resistance genes, 183 (93.4%) ESBL-producing *E. coli* isolates carried *bla*_CTXM_ genes, with *bla*_CTXM_ group 1 being the most frequently found group [55]. Thus, the results were similar to those of the present study, in which *bla*_CTX-M_ genes were detected most frequently, too. Comparing another French study from 2020, which tested *E. coli* strains from calves–isolated from the dominant flora for each sample (n = 280)–the results showed that the ESBL phenotype was identified for 2% of the *E. coli* strains. In this case, the proportions of co-resistance were high for streptomycin (>85%), tetracycline (82.5%), and kanamycin (72.5%), but lower for gentamicin (10%) and enrofloxacin (15%) [56]. No information on the phylogenetic groups was provided in the studies [54,55,56]. 

Concerning tetracycline genes in this work, in which *tet*(A) occurred most frequently (59.1%), there are only little data for Austria or neighboring countries such as Germany. An Austrian study from 2020 and 2022, which investigated the prevalence of ESBL-producing *E. coli* in dairy farms in fecal samples from calves and cows, showed that the most frequent resistance among all isolates was determined to be tetracyclines [54]. However, more detailed information on tetracycline genes was not reported. Comparing this study [54] with the one from Lower Austria, it should be mentioned that they also sampled cows and not only sick calves as in the one from Lower Austria. More surveys in this regard have been performed in the USA: Comparing an American study from Kentucky that cultured fecal samples for the detection of tetracycline-resistant and ESBL-producing *E. coli*, a total of 329 tetracycline-resistant *E. coli* isolates were detected, and all of them carried *tet*(A) and *tet*(B) either alone (97%) or together (3%) [57]. Another American study—also from Kentucky—examined fecal samples from feedlot cattle that were continuously fed rations with or without tylosin for the concentration and prevalence of tetracycline-resistant and ESBL-producing *E. coli*. Overall, 98% of the tetracycline-resistant *E. coli* isolates (n = 511) were positive for *tet*(A) alone (45.4%), *tet*(B) alone (46.3%), or *tet*(A) and *tet*(B) in common (6.2%), with the remaining 1.6% being positive for *tet*(C) [58]. The limiting factor in the comparison of this study is that the cattle were fed with or without antibiotics in the American study while none of the calves sampled in the Lower Austrian study had been fed antibiotics. In another study from Northern California, the prevalence of *E. coli* and *Enterococcus* spp. in the fecal samples of beef cattle at different life stages was investigated, with a total of 244 *E. coli* isolates detected [59]. It was found that the percentage of non-susceptible *E. coli* isolates by antimicrobial for tetracycline was 13.1% (32/244). Unfortunately, no information on the genes was provided. 

The VAGs *papC*, *fimH,* and *iucD*, which were detected in the *E. coli* isolates in this study, characterize the so-called UPEC pathotype (uropathogenic *E*. *coli*), which typically cause urinary tract infections [60] and is categorized as an extra-intestinal pathogen [61]. It is also possible that certain UPEC strains have virulence characteristics of diarrheagenic *E. coli* (DEC) pathotypes, which are usually associated with the enteroaggregative *E. coli* (EAEC) pathotype. Thus, it is possible that some fecal EAEC strains could be potential uropathogens and certain UPEC strains have acquired EAEC characteristics, becoming a potential cause of diarrheal disease [61]. A Russian study investigated the occurrence of 22 virulence-associated genes (VAGs) among 49 *E. coli* strains isolated from healthy cattle. It was found that ExPEC strains were the most common, as they were found in 55.1% of the studied strains, with only one strain having a high uropathogenic potential. A total of 17 strains (34.7%), however, contained genes associated with DEC pathotypes [62]. The following virulence factors of the ExPEC pathotype were detected in the Russian study in cows and calves, including *papC* with 20.4% and *fimH* with 91.8% [62]. However, this study did not establish a connection between the assumption that some fecal EAEC strains may be potential uropathogens, but also certain UPEC strains may have acquired EAEC properties and become a potential cause of diarrheal disease. 

From the samples in this study, the one MRSA strain isolated from the nasal swabs belonged to the *spa* type t011 and CC398; no MRSA was detected from the milk samples from cows. CC398 is not highly pathogenic in humans and is also associated with professional exposure to livestock animals [63]. The so-called livestock-associated (LA) MRSA-CC398 lineage is known as colonizers of livestock animals, for frequent multi-resistance to antimicrobials and its low host specificity [64,65]. In recent years, LA-MRSA has attracted much attention in both veterinary and human medicine [43,66]. Even though no MRSA was found in the milk samples from the Lower Austrian study, it cannot be generally assumed that (LA-) MRSA is not present in these cow stables, although the data for Austria in this regard are scarce. The following studies demonstrate that MRSA does play a major and significant role in dairy farming [67,68,69,70,71,72,73,74]. Further comparisons with the studies mentioned concerning the microbiological methods cannot be made, as no MRSA was detected in the milk samples from Lower Austria, and therefore no further microbiological tests were performed. In neighboring Germany, an investigation identified and characterized LA-MRSA in a collection of Staphylococci isolated from milk samples of cows (n = 14,924) and a CC398-specific PCR was performed for all *S. aureus* isolates [67]. A total of 327 *S. aureus* isolates were detected, of which 214 were epidemiologically independent and 12 were positive for LA-MRSA and carried the *mecA* gene [67]. A Dutch study investigated MRSA isolates from dairy farms and analyzed them for their genetic relatedness and antimicrobial susceptibility [67]. Among others, 46 MRSA isolates from 1389 milk samples were included in that study and all of them were positive for the clonal complex CC398 [68]. A Czech study also evaluated the diversity and molecular characteristics of MRSA in livestock. A total of 757 MRSA strains were analyzed, of which 34 (18%) originated from cattle. The most common *spa* type was t011, and 18 *mecA* positive strains were detected [69]. The presence of MRSA strains in bovine milk has been also reported in various other countries, including Italy, England, Turkey, and Greece [70,71,72,73,74], but the Austrian literature is limited in this respect. In Europe, to date, the most prevalent livestock-associated lineage of MRSA is CC398 [75,76]. In this study, the MRSA belonged to CC398 and carried SCC*mec* type IV. CC398-MRSA-IV is relatively uncommon within LA-MRSA [76] and is frequently detected in horses [77,78,79]. The finding that CC398-MRSA-IV was isolated from the one nasal swab should be mentioned, as it is relatively uncommon in LA-MRSA, but other studies have also reported the presence of CC398-MRSA-IV in bovines [80,81]. CC398 harboring SCC*mec* V is much more common, being observed in pigs, humans, and cattle [64,66,67]. Studies from Belgium, for example, that also have tested bovine nasal swabs for MRSA, have shown that SCC*mec* type IV has already been isolated in bovines, even if only rarely [80,81]. The fact that MRSA was found in 1 of the 123 nasal swabs from calves is quite surprising because all of the animals tested received antibiotic therapy when the calves suffered from pneumonia (especially tetracycline 50.1% and chloramphenicol 35.7%). This being a proven risk factor for MRSA colonization [82], which of course results from the (frequent) treatment with antibiotics. For example, a previous Dutch study also showed that calves that had received antibiotic therapy were more likely to be carriers of MRSA [83]. They collected 2151 nasal swabs from calves, with the result that the prevalence of MRSA was 28% with the predominant *spa* type t011 (80%). Unfortunately, it is not evident from the examinations which antibiotic therapy the calves had received, and the results are therefore inconclusive in comparison with the Lower Austrian study regarding antibiotic therapy, in which the calves were mainly given tetracyclines (50.1%) and chloramphenicol (35.7%) for the treatment of pneumonia. As the calves had received different antibiotics during one treatment, it was not possible to unravel the effect of the antibiotic classes or individual antibiotics. It was also found that MRSA transmission with age is higher in calves treated with antibiotics (*p* = 0.05) than in untreated animals, which means that older calves were more often MRSA positive than calves of younger age (OR = 1.3 (per 10 weeks)) [83]. A comparable study was conducted in Belgium, in which the nasal swabs from cattle were analyzed. The investigation also examined nasal swabs from cattle with the result that the prevalence of MRSA isolates was 19.8% [80]. A total of 88 MRSA isolates were recovered, of which 81 (96.3%) were positive in the CC398 PCR. When identifying the *spa* types, 64 (79%) were assigned to *spa* type t011. Other *spa* types recovered were t037 (n = 1), t121 (n = 1), t388 (n = 1), t1451 (n = 3), t1456 (n = 3), t1985 (n = 4), t3423 (n = 1), t6228 (n = 2), and a non-typeable *spa* type. The antibiotic resistances were as follows: Resistance to tetracycline (96.3%), trimethoprim (95.1%), clindamycin (86.4%), erythromycin (86.4%), kanamycin (80.2%), and gentamicin (76.5%) [80].

## 5. Conclusions

Currently, there is a paucity of Austrian data on the presence of extended-spectrum cephalosporin-resistant *E. coli* and MRSA in calves and cows. The findings of the present study demonstrate that AMR does indeed play a role in Austrian (or Lower Austrian) bovine stables, which is finally relevant for successful antibiotic therapy in sick cattle. 

Considering the close contact between farmers, veterinarians, and cattle, the findings of the present study highlight a significant public health concern. The identification of ESBL and MRSA underscores the potential for anthropozoonotic and zoonotic transmission. Continued surveillance of this issue is crucial.

## Figures and Tables

**Figure 1 animals-14-03383-f001:**
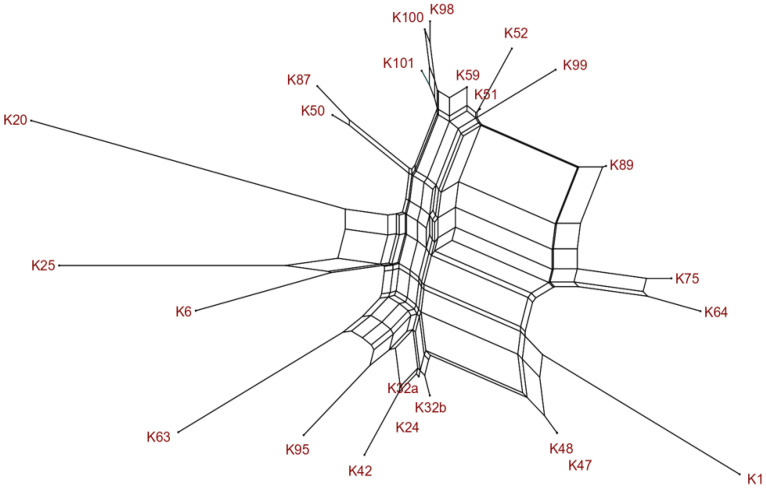
SplitsTree network of extended-spectrum cephalosporin-resistant *Escherichia coli*. Abbreviations: K, fecal sample.

**Table 1 animals-14-03383-t001:** Pheno- and genotypic characterization of *E. coli* isolated from calves.

Sample Number	Resistance Phenotype	Genotype	Phylotype	Virulence Associated Genes
K32a *	ESBL, TET, SXT	*bla*_CTX-M-1/15_, *bla*_TEM_, *tet*(A), *dfrA1*, *sul2*, *aadA1*, *aadA2*	B1	*fimH1*, *fimH2*
K59	ESBL, SXT	*bla*_CTX-M-1/15_, *bla*_TEM_, *dfrA14*, *sul2*	A	*fimH1*, *fimH2*
K32b *	ESBL, TET, SXT	*bla*_CTX-M1/15_, *aadA1*, *aadA2*, *tet*(A), *sul2*, *dfrA1*	B1	*fimH1*, *fimH2,*
K63 *	ESBL, CIP, GEN, TET, SXT, CHL	*bla*_CTX-M1/15_, *bla*_TEM_, *aadA1*, *aadA2*, *aphA*, *tet*(A), *sul2*, *sul3*, *dfrA12, cmlA1*, *floR*	A	*fimH1*
K42 *	ESBL, TET, SXT	*bla*_CTX-M1/15_, *bla*_TEM_, *aadA1*, *aadA2, tet*(A), *sul2*, *dfrA1*	B1	*fimH1*, *fimH2*, *papC1*, *papC2*, *iucD1*, *iucD2*
K64 *	ESBL, TET, SXT	*bla*_CTX-M1/15_, *aadA4*, *tet*(A), *sul2*, *dfrA17*	B1	*fimH1*, *fimH2*, *papC1*, *papC2*, *iucD1*, *iucD2*
K47 *	ESBL, TET, SXT	*bla*_CTX-M1/15_, *bla*_TEM_, *aadA1*, *aadA2, tet*(A), *sul2*, *dfrA1*, *dfrA5*	B1	*fimH1*, *fimH2*, *papC1*, *papC2*, *iucD1*, *iucD2*
K75 *	ESBL, CIP, TET, SXT, CHL	*bla*_CTX-M1/15_*, bla*_TEM_, *aadA4, tet*(A), *tet*(B), *dfrA17, cat*	B1	*fimH1*, *fimH2*, *papC1*, *papC2*, *iucD1*, *iucD2*
K48 *	ESBL, TET, SXT	*bla*_CTX-M1/15_, *bla*_TEM_, *aadA1*, *aadA2, tet*(A), *sul2*, *dfrA1*, *dfrA5*	B1	*fimH1*, *fimH2*, *papC1*, *papC2*, *iucD1*, *iucD2*
K89	ESBL, SXT	*bla*_CTX-M1/15_, *bla*_TEM_, *sul2*, *dfrA14*	A	*fimH1*, *fimH2*
K50	ESBL	*bla*_CTX-M9_, *bla*_TEM_, *dfrA14*	E clades	*fimH1*, *fimH2*
K95 *	ESBL, CIP, SXT, CHL	*bla*_CTX-M1/15_, *aadA1*, *aadA2*, *tet*(A), *sul3*, *dfrA12*, *cmlA1*	B1	*fimH1*, *hlyA-var2*
K51 *	ESBL, CIP, TET, SXT, CHL	*bla*_CTX-M1/15_, *bla*_TEM_, *tet*(A), *sul2*, *dfrA14*, *floR*	A	*fimH1*, *fimH2*
K99	ESBL, SXT	*bla*_CTX-M1/15_, *bla*_TEM_, *sul2*, *dfrA14*	A	*fimH1*, *fimH2*, *papC1*, *papC2*, *iucD1*, *iucD2*
K52	ESBL, SXT	*bla*_CTX-M1/15_, *bla*_TEM_, *sul2*, *dfrA14*	A	*fimH1*, *fimH2*
K1 *	ESBL, CIP, GEN, TET, SXT, CHL	*bla*_CTX-M1/15_, *aadA1*, *aadA2*, *aphA*, *tet*(A), *sul1*, *dfrA1*, *dfrA5*	A	*fimH1*, *hlyA-var2*
K6 *	AmpC, ESBL, CIP, GEN, TET, SXT, CHL	*bla*_CTX-M1/15_, *bla*_TEM_, *bla*_ACT_, *tet*(A), *tet*(B), *sul1*, *sul2*, *dfrA5*, *dfrA7*, *dfrA17*	B1	*fimH1*, *fimH2*
K24 *	ESBL, TET, SXT	*bla*_CTX-M1/15_, *bla*_TEM_, *aadA1*, *aadA2*, *tet*(A), *sul2*, *dfrA1*	B1	*fimH1*, *fimH2*
K87 *	ESBL, CIP, TET, SXT, CHL	*bla*_CTX-M9_, *bla*_TEM_, *dfrA14*, *floR*	A	*fimH1*, *fimH2*
K98	ESBL, SXT	*bla*_CTX-M1/15_, *bla*_TEM_, *sul2*, *dfrA14*	A	*fimH1*, *fimH2*
K100	ESBL, SXT	*bla*_CTX-M1/15_, *bla*_TEM_, *sul2*, *dfrA5*, *dfrA14*	A	*fimH1*, *fimH2*
K101	ESBL, SXT	*bla*_CTX-M1/15_, *bla*_TEM_, *sul2*, *dfrA5*, *dfrA14*	A	*fimH1*, *fimH2*

Abbreviations: CHL, chloramphenicol; CIP, ciprofloxacin; ESBL, Extended-spectrum lactamase phenotype, i.e., resistance to penicillins and cephalosporins of first to third generations; GEN, gentamicin; TET, tetracycline; SXT, trimethoprim–sulfamethoxazole; * MDR, multidrug-resistant.

## Data Availability

The original contributions presented in the study are included in the article/Appendix A, further inquiries can be directed to the corresponding author.

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
