# Peer review of "Characterizing Methicillin-Resistant Staphylococcus spp. and Extended-Spectrum Cephalosporin-Resistant Escherichia coli in Cattle"

_animals, 2024, doi:10.3390/ani14233383_

Round 1

Reviewer 1 Report

Comments and Suggestions for Authors

See attached file

Author Response

Comment 1: L36: in the abstract replace this sentence « Twenty-two Escherichia coli isolates were detected among 36 fecal samples » after the paragraph concerning S. aureus, just before L38 « All of the Escherichia coli isolates were resistant to ceftazidime » to be more coherent.

Response 1: We followed the recommendation of this reviewer:

No methicillin-resistant Staphylococcus aureus was found in the milk samples, and one nasal swab was positive for methicillin-resistant Staphylococcus aureus. Twenty-two Escherichia coli isolates were detected among fecal samples. All of the Escherichia coli isolates were resistant to ceftazidime.

[Page 1, Abstract, Lines 45,46]

Comment 2: L50-51: Remove (S) and (E) from the sentence.

Response 2:  Thank you, corrected:

In veterinary medicine, methicillin-resistant Staphylococcus aureus (MRSA) and extended-spectrum β-lactamase (ESBL)-producing Escherichia coli are considered pathogens that are difficult to treat due to increasing antimicrobial resistance (AMR).

[Page 2, Introduction, Lines 58,59]

Comment 3: L65 : ExPEC not EXPEC

Response 3: Thank you, corrected:

Extraintestinal infections are caused by extraintestinal pathogenic E. coli strains (ExPEC). ExPEC are mostly harmless intestinal commensals and are only harmful if they reach or got dislocated to other parts of the body [7-9].

[Page 2, Introduction, Line 72]

Comment 4: L76: Resistance to beta-lactam antibiotics is caused mainly by PBP2a: this latter is not the only cause of B lactam resistance, there is also penicillin binding protein 2c, especially since you talk about the mecC gene after. I suggest « Resistance to beta-lactam antibiotics is caused by a modified penicillin-binding protein PLP2a or PLP2c- which have only a low affinity for beta-lactams and involve respectively mecA and mecC genes »

Response 4: Thank you for your comment. PBP2a and PBP2c are classified as the penicillin-binding proteins, and they are responsible for bacterial resistance to beta-lactam antibiotics.

Resistance to beta-lactam antibiotics is caused by a modified penicillin-binding protein PBP2a or PBP2c- which have only a low affinity for beta-lactams and involve respectively mecA and mecC genes.

[Page 3, Introduction, Lines 106-109]

Comment 5: I suggest that the sentences 93-94 should be placed in line 72 just after « which are frequently associated with opportunistic infections »

Response 5: Thank you, corrected:

Staphylococci are part of the physiological microbiota of the skin and mucous membranes of humans and different animals, and they are frequently associated with opportunistic infections [16,17]. In cows, S. aureus is a very relevant pathogen in udder infections, but it can also be found on the skin and mucous membranes, such as the nasopharynx [18-21].

[Page 3, Introduction, Lines 101-103]

Comment 6: L98: you should add the title 1.2 Sample collection

Response 6: Thank you, corrected:

2.1. Sample collection

[Page 3, 2. Materials and Methods, Line 136]

Comment 7: L99-101: « All samples were collected as part of veterinary herd management, which is why no authorization from the Ethics and Animal Welfare Committee of the University of Veterinary Medicine Vienna was required » should be placed in the Ethics statement in the end of the manuscript not in material and methods

Response 7: We followed the recommendation of this reviewer:

Ethic Statement: All samples were collected as part of veterinary herd management, so no authorization from the Ethics and Animal Welfare Committee of the University of Veterinary Medicine Vienna was required.

[Page 11, Ethic Statement, Lines 467-469]

Comment 8: L111: Please specify the morphotype, on which basis the selection of the colony was done.

Response 8: Thank you, corrected.

The selection of the colony was based on form (e.g. circular, irregular), elevation (e.g. flat, convex), margin (e.g. entire, undulate), and mucoid vs. non-mucoid.

[Page 3, 2.1. Sample collection, Lines 142-144]

Comment 9: L116: Specify the medium used for the antibiogram (Mueller-Hinton?).

Response 9: Thank you for this comment. Please refer to the CLSI standards for exact details.

Antimicrobial susceptibility testing was performed by agar disk diffusion according to the CLSI standards [31].

[Page 3, 2.1. Sample Collection, Line 147,148]

Comment 10: L141: Glycopetids (vancomycin) were not tested? If not, it is important to test since vancomycin resistance is frequent in MRSA strains

Response 10: Thank you for this comment. Vancomycin was not tested because there are no clinical breakpoints (CBP) for the combination of S. aureus and vancomycin according to CLSI standards [31]. MIC was not performed during the present study.

Comment 11: In the material and methods, no data concerning the milk samples were not mentioned. Besides, all genes tested by PCR should be mentioned (bla…)

Response 11: Thank you. We have added the methods for the milk samples. Resistance genes were analyzed by PCRs (i.e., catA, cmlA, floR, tet(A), tet(B)), and VF was detected by microarray-based assays.

The milk samples were cultivated and identified as previously described by Keinprecht et al. [37].

[Page 4, 2.2. Isolation of MRSA, antimicrobial susceptibility testing, Lines 175,176]

Resistance genes were analyzed by PCRs (i.e.; catA; cmlA; floR; tet(A), tet(B)) as described elsewhere [32].

[Page 4, 2.3. Detection of Antimicrobial Resistance Genes and Molecular Characterization of MRSA, Lines 187,188]

Comment 12: L154-158: « Out of the 412 samples, 190 (46.1%) were from cows suffering from mastitis, 123 (29.9%) from pneumonia calves and 99 (24.0%) from diarrhea calves. No positive sample/isolate was detected in the 123 milk samples. Of the 123 calves suffering from pneumonia, one (0.8%) positive sample/isolate was found » This paragraph doesn’t correspond to the title of the paragraph « Antimicrobial Susceptibility Testing and Characterization of Genotypic Antibiotic Resistance of E. coli » which concerns only E coli isolates. So, sentences concerning nasal and milk samples should be placed in paragraph 3.3 of the results.

Response 12: We followed the recommendation of this reviewer:

3.3. MRSA Characterization

Out of the 412 samples, 190 (46.1%) were from cows suffering from mastitis, and 123 (29.9%) were from pneumonia calves. No positive sample was detected in the 123 milk samples. Of the 123 calves suffering from pneumonia, one (0.8%) MRSA isolate was found.

[Page 7, 3.3. MRSA Characterization, Lines 243-245]

Comment 13: L 161: How can you say that EBSL phenotype is detected among isolates while G4G were not tested. Also, EBSL genes are cefoxitin sensible.

Response 13: Thank you for this comment. Please specify what you mean by the abbreviation G4G so that we can edit this comment.

Comment 14: You didn’t find any E. coli isolates harboring AmpC gene?

Response 14: Thank you for this comment. We found one E. coli isolate that harbored an AmpC gene (blaACT) and showed a reduction in the inhibition zone of cefoxitin (30 µg). We have updated this in the manuscript.

[Simple Summary Lines 26,27; Abstract Line 49; Introduction Lines 89-98; Results Lines 219-221; Discussion Lines 309-321]

Comment 15: The discussion is only a listing of results of different publications without trying to explain the results found in your study or the differences between the studies

Response 15: We followed the recommendation of this reviewer.

[Pages 7-11]

Comment 16: Concerning VAGs papC, fimH, and iucD genes detected in your study, you mention in the discussion UPEC strains can have virulence characteristics of diarrheagenic E. coli (DEC) pathotypes (L302), but you didn’t compare your results to others studies.

Response 16: Thank you, corrected.

A Russian study investigated the occurrence of 22 virulence-associated genes (VAGs) among 49 E. coli strains isolated from healthy cattle. It was found that ExPEC strains were the most common, as they were found in 55.1 % of the studied strains, with only one strain having a high uropathogenic potential. Seventeen strains (34.7%), though, contained genes associated with DEC pathotypes [63]. The following virulence factors of the ExPEC pathotype were detected in the Russian study in cows and calves, including papC with 20.4% and fimH with 91.8% [63].  However, this study did not establish a connection between the assumption that some fecal EAEC strains may be potential uropathogens, but also certain UPEC strains may have acquired EAEC properties and become a potential cause of diarrheal disease.

[Page 9 and 10, Discussion, Lines 379-389]

Response to Comments on the Quality of English Language

Point 1:

Response 1: We made changes and modifications in the English language.

Reviewer 2 Report

Comments and Suggestions for Authors

Comments to the Author

The authors have conducted interesting and informative research that can add useful information to the existing data on methicillin-resistant S. aureus and broad-spectrum cephalosporin-resistant E. coli in cattle. However, my major concern is that more details are required in the methods section.  The methods are sufficiently described to allow the study to be repeated. Statistics data analysis is not found in the manuscript.

1- Please adjust the footer "Animals 2024"

2-Abstract at Lines 30-32 "milk and nasal samples were examined for the presence of methicillin-resistant Staphylococcus aureus and fecal samples for extended-spectrum cephalosporin-resistant Escherichia coli." The authors specified the samples however in the results section they mentioned the results of the isolation of E. coli from all sample types. Also please revise Lines 21-22 in summary.

3- In keywords, please add "mastitis, pneumonia, diarrhea, and virulence genes"

4- Line 132 "The nasal swabs were incubated overnight in tryptic soy broth" What about milk samples for isolation of S. aureus?

5- Lines 153, 154: italicize "E. coli" and "bla"  and the other genes throughout the manuscript

6- "lines 156-158: E. coli was not isolated from mastitis, but were isolated from pneumonia, and diarrhea however in the methods section authors write only isolation from fecal samples (L107). please revise this point.

7-line 156 "No positive sample/isolate was detected in the 123 milk samples.": 190 instead of 123 

8- lines 157-158 "Of the 123 calves suffering from pneumonia, one (0.8%) positive sample/isolate was found. Of the 99 calves suffering from diarrhea, 21 (21.2%) positive samples and 22 (22.2%) isolates were detected.": please revise the total Nos of E. coli isolates it should be 23 not 22.

9-Table 1: the title included E. coli so please delete the column "species" and replace "phenotype" with "resistance phenotype" and write the other  antibiotics to which isolates were resistant to it "MDR isolates" 

10-please write more details in the methodology section for Genotyping by Inter-Array Kit CarbaResist and for the microarray hybridization kit used for the DNA-based detection of virulence genes, pathogenicity markers of S. aureus, and antibiotic resistance genes, spa typing. 

11- Line  164"The most frequent combination of multidrug-resistant phenotype  [35]" why this reference is mentioned in the results.

12-190 sterile milk samples: delete the word "sterile"

13-Table 2. not required authors could write data in the text.

Author Response

1. Summary

Thank you very much for taking the time to review this manuscript. Please find the detailed responses below and the corresponding and corrected revisions in the re-submitted files.

2. Point-by-point response to Comments

Review 2:

Comment 1: Please adjust the footer "Animals 2024"

Response 1: Thank you, corrected.

Comment 2: Abstract at Lines 30-32 "milk and nasal samples were examined for the presence of methicillin-resistant Staphylococcus aureus and fecal samples for extended-spectrum cephalosporin-resistant Escherichia coli." The authors specified the samples however in the results section they mentioned the results of the isolation of E. coli from all sample types. Also please revise Lines 21-22 in summary.

Response 2: Thank you for the comment. The mistake was in the results section "3.1 Antimicrobial Susceptibility Testing and Characterization of Genotypic Antibiotic Resistance of E. coli". Here we have deleted the passage for milk samples and nasal swabs and added it to “3.3. MRSA Characterization”. Lines 21-22 and 30-32 should now also be correct.

3.3. MRSA Characterization

Out of the 412 samples, 190 (46.1%) were from cows suffering from mastitis, and 123 (29.9%) were from pneumonia calves. No positive sample was detected in the 190 milk samples. Of the 123 calves suffering from pneumonia, one (0.8%) MRSA isolate was found.

[Page 7, 3.3. MRSA Characterization, Lines 243-245]

Comment 3: In keywords, please add "mastitis, pneumonia, diarrhea, and virulence genes"

Response 3: Thank you, corrected.

Keywords: MRSA; ESBL; cattle; bovine; E. coli; AMR, mastitis, pneumonia, diarrhea, virulence genes

[Page 2, Keywords, Lines 54,55]

Comment 4: Line 132 "The nasal swabs were incubated overnight in tryptic soy broth" What about milk samples for isolation of S. aureus?

Response 4: Thank you, corrected.

The milk samples were cultivated and identified as previously described by Keinprecht et al. [37].

[Page 4, 2.2. Isolation of MRSA, antimicrobial susceptibility testing, Lines 175,176]

Comment 5: Lines 153, 154: italicize "E. coli" and "bla" and the other genes throughout the manuscript

Response 5: Thank you, corrected.

Comment 6: "Lines 156-158: E. coli was not isolated from mastitis, but were isolated from pneumonia, and diarrhea however in the methods section authors write only isolation from fecal samples (L107). please revise this point.

Response 6: Thank you for this comment. We isolated E. coli only from fecal samples.

Comment 7: line 156 "No positive sample/isolate was detected in the 123 milk samples.": 190 instead of 123

Response 7: Thank you, corrected.

No positive sample/isolate was detected in the 190 milk samples.

[Page 7, 3.3. MRSA Characterization, Lines 244,245]

Comment 8: lines 157-158 "Of the 123 calves suffering from pneumonia, one (0.8%) positive sample/isolate was found. Of the 99 calves suffering from diarrhea, 21 (21.2%) positive samples and 22 (22.2%) isolates were detected.": Please revise the total Nos of E. coli isolates it should be 23 not 22.

Response 8: Thank you for this comment. Out of 99 fecal samples, 21 positive samples were detected, from which 22 extended cephalosporin-resistant E. coli could be isolated.

Comment 9: the title included E. coli so please delete the column "species" and replace "phenotype" with "resistance phenotype" and write the other antibiotics to which isolates were resistant to it "MDR isolates"  

Response 9: Thank you, corrected.

[Page 5 and 6, Table 1]

Comment 10: please write more details in the methodology section for Genotyping by Inter-Array Kit CarbaResist and for the microarray hybridization kit used for the DNA-based detection of virulence genes, pathogenicity markers of S. aureus, and antibiotic resistance genes, spa typing.

Response 10: Thank you for this comment. We consider that the detailed description of Inter-Array Kit CarbaResist, the Microarray Hybridization Kit, and spa typing is beyond the scope of this manuscript. Therefore, we kindly ask you to read the details from the cited references.

Microarray-Based Detection of Virulence-Associated Genes was performed as described by Bernreiter-Hofer et al. [38]. Genotyping by Inter-Array Kit CarbaResist and the Microarray Hybridization Kit were performed as described by Bedenic et al, [39], Braun et al. [40-42], and Monecke et al. [43]. DNA extraction was as previously described by Loncaric et al. [44]. Spa typing was performed as previously described by Loncaric et al. [44].

Resistance genes were analyzed by PCRs (i.e.; catA; cmlA; floR; tet(A), tet(B)) as described elsewhere [32].

[Page 4, 2.3. Detection of Antimicrobial Resistance Genes and Molecular Characterization of MRSA, Lines 181-188]

Comment 11: Line 164"The most frequent combination of multidrug-resistant phenotype [35]" why this reference is mentioned in the results.

Response 11: Thank you for this comment. Reference number 35 (now reference number 45) is cited here to refer to the definition of multidrug resistance (MDR). However, since the reference is already mentioned in the previous sentence and thus refers to the definition of MDR, so we deleted it here.

Comment 12: 190 sterile milk samples: delete the word "sterile"

Response 12: Thank you, corrected.

No positive sample/isolate was detected in the 190 milk samples.

[Lines 21, 36, 391]

Comment 13: Table 2. not required authors could write data in the text.

Response 13: Thank you, corrected.

[Page 7, 3.3. MRSA Characterization, Lines 246-252]

Response to Comments on the Quality of English Language

Point 1:

Response 1: We made changes and modifications in the English language.

Reviewer 3 Report

Comments and Suggestions for Authors

 Presence and characterization of methicillin-resistant Staphylococcus  spp. (MRS) and extended-spectrum cephalosporin-resistant Escherichia coli in cattle - a pilot study

The article Presence and characterization of methicillin-resistant Staphylococcus spp. (MRS) and extended-spectrum cephalosporin-resistant Escherichia coli in cattle - a pilot study provides important information; however, it needs modifications and improvements in the English language before it can be published.

I suggests change the title to: Characterizing Methicillin-Resistant Staphylococcus spp. and Extended-Spectrum Cephalosporin-Resistant Escherichia coli in Cattle.

5 out of 72 citations are from the authors themselves; however, I consider these citations appropriate in the sections where they were made.

Line 21 – Sterile Milk? Did you try to isolate bacteria from sterile milk?

Line 27 - The simple summary ended abruptly.

Line 33 – Milk isn’t sterile.

Line 35 – What kind of treatment?

Line 36 – Need to add a short resume of the methodology in the abstract.

Line 55 – Rewrite: E. coli is a multifaceted bacterium and part of the normal microbiota but can also play an important role as pathogen.

Line 66-68 – Confuse, need to be rewritten.

Regarding E. coli, no molecular mechanism associated with antimicrobial resistance was reported in the introduction, whereas it was provided for Staphylococci. I suggest including the same level of detail for E. coli

Lines 93-94 – This sentence seems kind of lost in the section.

Line 95-97 – English needs to be improved.

Line 153-154 –  Italic in E. coli is missing.

Line 153-159 – Need to improve this sentence. The same information appears twice.

Line 163 –170 - I suggest rewriting all these sentences. It is confusing.

Line 198 - Sterile Milk? Did you try to isolate bacteria from sterile milk?

Line 209 – No methodology about MLST was described in material and methods section.

Comments on the Quality of English Language

Need to be improved. 

Author Response

1. Summary

Thank you very much for taking the time to review this manuscript. Please find the detailed responses below and the corresponding and corrected revisions in the re-submitted files.

2. Point-by-point response to Comments

Review 3:

Comment 1: The article Presence and characterization of methicillin-resistant Staphylococcus spp. (MRS) and extended-spectrum cephalosporin-resistant Escherichia coli in cattle - a pilot study provides important information; however, it needs modifications and improvements in the English language before it can be published.

I suggest changing the title to: Characterizing Methicillin-Resistant Staphylococcus spp. and Extended-Spectrum Cephalosporin-Resistant Escherichia coli in Cattle.

Response 1: We followed the recommendation of this reviewer:

Characterizing Methicillin-Resistant Staphylococcus spp. and Extended-Spectrum Cephalosporin-Resistant Escherichia coli in Cattle.

[Page 1, Titel, Lines 3, 4]

Comment 2: 5 out of 72 citations are from the authors themselves; however, I consider these citations appropriate in the sections where they were made.

Response 2: Thank you.

Comment 3: Line 21 – Sterile Milk? Did you try to isolate bacteria from sterile milk?

Response 3: Thank you, corrected.

A total of 190 milk samples from cows and 123 nasal swabs from cattle were examined for the presence of MRSA and 99 bovine fecal swabs for E. coli.

[Page 1, Simple Summary, Line 21]

Comment 4: Line 27 - The simple summary ended abruptly.

Response 4: Thank you, corrected:

Finally, the isolates were analyzed for the microbiological tests: DNA microarray, PCRs, and spa typing were conducted. The results conclusively showed that antibiotic resistance does play a role in cattle in (Lower) Austria.

[Page 1, Simple Summary, Lines 28-30]

Comment 5: Line 33 – Milk isn’t sterile.

Response 5: Thank you, corrected.

For milk samples, the first milk jets were milked into a pre-milking cup and then the teats were cleaned and disinfected and the samples were taken.

[Page 1, Abstract, Line 36]

Comment 6: Line 35 – What kind of treatment?

Response 6: Thank you for this comment. A distinction must be made between acute (with fever) and subclinical (without fever) mastitis. A milk sample was taken for each udder inflammation. If the cow had a poor general condition with fever (acute mastitis), antibiotic therapy with NSAIDs and, if necessary, infusions was started. If necessary, the therapy was adjusted according to the antibiogram. If the cow was in good general condition (subclinical mastitis), treatment was started with NSAIDs, and antibiotic therapy was started after the antibiogram.

Depending on the severity of the mastitis (acute mastitis or subclinical mastitis), antibiotics and non-steroidal anti-inflammatory drugs were given immediately (acute disease) or after completion of the antibiogram (subclinical disease).

[Page 1, Abstract, Lines 39-41]

Comment 7: Line 36 – Need to add a short resume of the methodology in the abstract.

Response 7: Thank you, corrected:

Isolates were characterized by a polyphasic approach including susceptibility pheno- and genotyping and microarray-based assays.

[Page 1, Abstract, Lines 42,43]

Comment 8: Line 55 – Rewrite: E. coli is a multifaceted bacterium and part of the normal microbiota but can also play an important role as pathogen.

Response 8: We followed the recommendation of this reviewer:

E. coli is a multifaceted bacterium and part of the normal microbiota but can also play an important role as a pathogen.

[Page 2, Introduction, Lines 63,64]

Comment 9: Line 66-68 – Confuse, need to be rewritten.

Response 9: We followed the recommendation of this reviewer:

E. coli has acquired resistance mechanisms and a genetic adaptation to exposure to antibiotics. Due to this genetic adaptation, antimicrobials may be less effective against E. coli, which can result in reduced susceptibility to antimicrobials [10,11].

[Page 2, Introduction, Lines 73-76]

Comment 10: Regarding E. coli, no molecular mechanism associated with antimicrobial resistance was reported in the introduction, whereas it was provided for Staphylococci. I suggest including the same level of detail for E. coli

Response 10: We followed the recommendation of this reviewer:

E. coli can highly accumulate resistance genes, mostly through horizontal gene transfer, which means this bacterial species is no longer susceptible to antimicrobial agents. The most problematic mechanisms in E. coli correspond to the acquisition of genes encoding carbapenemases, extended-spectrum β-lactamases, plasmid-mediated quinolone resistance genes, 16S rRNA methylases, and mcr-genes [12]. ESBL genes among E. coli from animals are associated with several insertion sequences (ISs; ISEcp1, ISCR1, IS26, IS10) transposons (Tn2), and integrons [12,13]. Most ESBL genes are plasmid-located and the most prevalent replicon types identified among ESBL-carrying plasmids from E. coli: IncF, IncI1, IncN, IncHI1, and IncHI2 [73, 87]. Some plasmids harbor other resistance genes besides the ESBL gene, which may facilitate coselection and persistence of ESBL gene-carrying plasmids even without the selection pressure of β-lactams when the appropriate antimicrobial agents are used [12,14].

[Page 2, Introduction, Lines 76-88]

Comment 11: Lines 93-94 – This sentence seems kind of lost in the section.

Response 11: Thank you, corrected:

Staphylococci are part of the physiological microbiota of the skin and mucous membranes of humans and different animals, which are frequently associated with opportunistic infections [16,17]. In cows, S. aureus is a very relevant pathogen in udder infections but it also can be found on the skin and mucous membranes, such as the nasopharynx [18-21].

[Page 3, Introduction, Lines 101-103]

Comment 12: Lines 95-97 – English needs to be improved.

Response 12: Thank you, corrected:

The general aim of this study was to investigate the presence of ESBL-producing E. coli and MRSA in cattle in Austrian stables and to what extent AMR plays a role in the presence of these bacteria.

[Page 3, Introduction, Lines 124-126]

Comment 13: Line 153-154 – Italic in E. coli is missing.

Response 13: Thank you, corrected:

A total of 22 E. coli isolates were grown on MacConkey agar with cefotaxime (MacCTX) and identified as E. coli by MALDI-TOF MS.

[Page 5, 3.1. Antimicrobial Susceptibility Testing and Characterization of Genotypic Antibiotic Resistance of E. coli, Lines 193,194]

Comment 14: Line 153-159 – Need to improve this sentence. The same information appears twice.

Response 14: We followed the recommendation of this reviewer:

A total of 22 E. coli isolates were grown on MacConkey agar with cefotaxime (MacCTX) and identified as E. coli by MALDI-TOF MS. Out of the 412 samples, 99 (24.0 %) came from calves with diarrhea. Of these, 21 (21.2 %) positive samples and 22 (22.2 %) isolates were detected.

[Page 4, 3.1. Antimicrobial Susceptibility Testing and Characterization of Genotypic Antibiotic Resistance of E. coli, Lines 193-199]

Comment 15: Lines 163 –170 - I suggest rewriting all these sentences. It is confusing.

Response 15: We followed the recommendation of this reviewer:

In total, 63.6% (n=14) of the isolates exhibited a multidrug-resistant (MDR) phenotype [35]. Resistance to β-lactams, trimethoprim-sulfamethoxazole (SXT), and tetracyclines were most common among the multidrug-resistant phenotypes (n=13, 59.1%). Further, the most frequently observed resistance property was combined resistance to β-lactams tested antibiotics and trimethoprim-sulfamethoxazole (n=21, 95.5%).  Concerning antibiotic resistance to non-β-lactam antibiotics, the following results were obtained: antibiotic resistance to trimethoprim-sulfamethoxazole (n=21, 95.5%), tetracyclines (n=13, 59.1%), ciprofloxacin (n=7, 31.8%), chloramphenicol (n=7, 31.8%) and gentamicin (n=3, 13.6%) was determined.

[Page 5, 3.1. Antimicrobial Susceptibility Testing and Characterization of Genotypic Antibiotic Resistance of E. coli, Lines 203-211]

Comment 16: Line 198 - Sterile Milk? Did you try to isolate bacteria from sterile milk?

Response 16: Thank you, corrected:

No positive sample was detected in the 190 milk samples.

[Page 7, 3.3. MRSA Characterization, Lines 244,245]

Comment 17: Line 209 – No methodology about MLST was described in material and methods section.

Response 17: No methodology for MLST was described in Material and Methods because no MLST was done.

Response to Comments on the Quality of English Language

Point 1:

Response 1: We made changes and modifications in the English language.

Round 2

Reviewer 2 Report

Comments and Suggestions for Authors

The authors have improved the manuscript. However, more details should be added to the methods section of "Detection of Antimicrobial Resistance Genes and Molecular Characterization of MRSA (lines 176-180)." Writing references is not sufficient.

Please italicize "Escherichia coli" on line 232. 

Author Response

1. Summary

Thank you very much for taking the time to review this manuscript again. Please find the detailed responses below and the corresponding and corrected revisions in the re-submitted files.

  1. Point-by-point response to Comments

Review 2:

Comment 1: The authors have improved the manuscript. However, more details should be added to the methods section of "Detection of Antimicrobial Resistance Genes and Molecular Characterization of MRSA (lines 176-180)." Writing references is not sufficient.

Response 1: Thank you for your comment. We have listed all target genes from InterArray as a supplement and submitted them online to provide more detailed information in this context.

Comment 2: Please italicize "Escherichia coli" on line 232.

Response 2: Thank you, corrected.

Figure 1. SplitsTree network of extended-spectrum cephalosporin-resistant Escherichia coli.

[Page 7, Figure 1, Line 231]

Reviewer 3 Report

Comments and Suggestions for Authors

 All suggestions have been addressed.

Author Response

Comment 1: All suggestions have been addressed.

Response 1: Thank you.